# Muscle Exercise Mitigates the Negative Influence of Low Socioeconomic Status on the Lack of Muscle Strength: A Cross-Sectional Study

**DOI:** 10.3390/healthcare9101244

**Published:** 2021-09-22

**Authors:** Hanna Lee, Mi-Ji Kim, Junhee Lee, Mingyo Kim, Young Sun Suh, Hyun-Ok Kim, Yun-Hong Cheon

**Affiliations:** 1Department of Internal Medicine, Division of Rheumatology, Gyeongsang National University Jinju Hospital, Jinju 52727, Korea; hanna890117@gmail.com (H.L.); mingyokim1@gmail.com (M.K.); 2Department of Internal Medicine, Division of Rheumatology, Gyeongsang National University School of Medicine, Jinju 52727, Korea; tatabox123@hanmail.net (Y.S.S.); galleonok@naver.com (H.-O.K.); 3Department of Preventive Medicine, Institute of Health Sciences, Gyeongsang National University College of Medicine, Jinju 52727, Korea; mijikim@gnuh.co.kr; 4Department of Health Informatics and Health Information Management, University of Washington, Seattle, WA 98195, USA; anakdl04@uw.edu; 5Department of Internal Medicine, Division of Rheumatology, Gyeongsang National University Changwon Hospital, Changwon 51472, Korea

**Keywords:** socioeconomic status, sarcopenia, handgrip strength, muscle mass, muscle exercise

## Abstract

Socioeconomic status (SES), which takes into account household income and education level, is an important factor in the role of muscle strength as a discriminator of sarcopenia. Although the benefits of exercise on muscle strength are well recognized, its influence on people of different SES has not been fully elucidated, informing the aim of this study. A total of 6081 subjects, for which we had complete data on measurements of handgrip strength (HGS) and other relevant variables, were included from the Korea National Health and Nutrition Examination Surveys (KNHANES) VII-3. A multivariable analysis showed that people with a low household income (odds ratio (OR) 1.637, *p* = 0.005) and low education status (OR 2.351, *p* < 0.001) had a poor HGS compared to those with a high SES, and that the difference in HGS made by muscle exercise was greater for people with a low household income (OR 7.082 vs. 3.619, *p* < 0.001) and low education status (OR 14.711 vs. 6.383, *p* < 0.001). Three-step logistic regression analysis showed that muscle exercise mediated the relationship between muscle strength and low household income (OR from 1.772 to 1.736, z = 2.373, *p* = 0.017) and low education level (OR from 2.368 to 2.309, z = 2.489, *p* = 0.012). This study confirmed that exercise improves the negative effect of SES on muscle strength, suggesting the greater importance of muscle exercise for people with a low SES.

## 1. Introduction

Sarcopenia is defined as a decrease in muscle mass and strength with aging. The prevalence of sarcopenia is increasing worldwide and in Korea, where the prevalence is exacerbated by the aging population. According to the Korea National Health and Nutrition Examination Survey (KNHANES (IV)), where sarcopenia was defined as appendicular skeletal muscle mass (ASM)/height^2^, the prevalence of sarcopenia was 31.2% in men and 8.8% in women over 65 years of age in Korea [1]. Sarcopenia can lead to obesity and visceral obesity, by reducing energy consumption by way of decreased physical activity, leading to diabetes (DM), hyperlipidemia, and hypertension (HTN) [1,2]. Additionally, it increases the risk of injury from falling by a factor of 1.52, which has negative health consequences, such as hospital admission, disability, and death [3,4,5], leading to an increase in medical costs. In 2000, the social cost of sarcopenia in the United States amounted to USD 18.5 billion [6].

According to the recommendations of the 2019 Asian Working Group for Sarcopenia (AWGS 2019), sarcopenia was diagnosed as a decrease in muscle mass, measured by dual-energy X-ray absorptiometry (DXA) (M < 7.0 kg/m^2^, F < 5.4 kg/m^2^) or bioelectrical impedance analysis (BIA) (M < 7.0 kg/m^2^, F < 5.7 kg/m^2^), and in muscle strength, measured by handgrip strength (HGS) (M < 28 kg, F < 18 kg) [7]. Although both muscle mass and strength measurements are required for the diagnosis of sarcopenia, many studies have shown that complications due to sarcopenia are more related to muscle strength than muscle mass. According to a cross-sectional study using data from the National Health and Nutrition Examination Survey (NHANES), lower leg muscle strength was associated with a degree of physical disability even after adjusting for age, sex, alcohol consumption, chronic disease, physical activity level, and smoking [8]. In a longitudinal study, the incidence of physical disability 2.64 times higher in men in the lowest quartile of knee extensor strength and 2.15 times higher in women in the lowest quartile of knee extensor strength [9]. In a study of 919 older women aged 65–101 years, individuals in the lowest tertile of grip strength were 2.17 times more likely to die from cardiovascular disease than those in the group with the highest tertile grip strength, even after adjusting for confounding factors [10]. In a study of 2292 older individuals aged 70–79 years, the risk of death increased for every 1 SD decrease in grip strength, even after adjusting for confounding factors, including muscle mass (by DXA) (male hazard ratio = 1.36, female hazard ratio = 1.67) [11]. Therefore, it appears that muscle strength predicts physical disability better than muscle mass, and its association with death is more robust.

HGS is a standard indicator of overall muscle strength and can be measured quickly, safely, and conveniently in older adults [12]. It is measured with the elbow extended as far as possible and, if necessary, with the elbow bent at a 90° angle while seated. Both hands or the predominant hand are measured at least twice, and the highest number is used [7]. Age and female sex are well known risk factors for poor HGS [13]. Low economic and education statuses are also known risk factors. Several studies have shown a difference in HGS according to SES [14,15]—specifically, low SES correlates with poor HGS. Therefore, many of the health problems caused by poor muscle strength become more of a problem with low SES. Exercise is the only known factor for improving muscle strength [13,16,17]; therefore, it becomes even more important for people of low SES. In addition, nutritional status, appropriate exercise and health awareness, lifestyle, and activity levels vary with SES. Thus, the effectiveness of exercise in improving muscle strength may be different depending on SES. However, no studies to date have evaluated this hypothesis.

This study was performed to assess whether there was a difference in improving muscle strength by exercise according to SES in the general Korean population.

## 2. Materials and Methods

### 2.1. Study Population

Data in this study were obtained from KNHANES VII (N = 24,269, performed from January 2016 to December 2018) (IRB 2018-01-03-P-A). KNHANES is a national survey conducted by the Korea Centers for Disease Control and Prevention to assess the health and nutritional status of Koreans. Each year, approximately 10,000 individuals are sampled as survey subjects to collect information on biochemical and clinical profiles for SES, health-related behaviors, quality of life, medicine use, anthropometrics, noncommunicable diseases, and diet. Health interviews and screenings were conducted at mobile screening centers by trained staff, including doctors, medical technicians, and health interviewers, and follow-ups were conducted by nutritionists while visiting the study participants’ homes. This cross-sectional survey is representative of the general Korean population [18]. We recruited 7992 participants of a KNHANES VII-3 survey and excluded a group of 1721 people who did not have HGS data and another group of 640 who were younger than the age of 20. The total number of participants of the cross-sectional study was 6081 (Figure 1).

### 2.2. Demographic Characteristics and Health-Related Variables

Information about sociodemographic characteristics was obtained through a health interview survey in the KNHANES. The sociodemographic characteristics were sex, age, education level (elementary school or less, middle school, high school, and college or more), and household income. Household income was divided into four groups: low (quartile one), low–medium (quartile two), high–medium (quartile three), and high (quartile four). 

Information about health-related characteristics was obtained through a health interview survey and an anthropometric survey in the KNHANES. The health-related characteristics considered were muscle and aerobic exercises, subjective health, and BMI (kg/m^2^). Muscle exercise was evaluated as the percentage of strength exercises, such as push-ups, sit-ups, dumbbells, weights, and iron bars, practiced for more than two days during the previous week. Participants were divided into two groups: exercise for less than 1 day a week and exercise for more than 2 days a week. The rate of aerobic physical activity was equivalent, for each activity performed, to medium-intensity physical activity for 2 h 30 min or more, high-intensity physical activity for 1 h 15 min or more, or a mixture of medium- and high-intensity physical activity (1 min for high intensity and 2 min for medium intensity). This was defined as the fraction of time practiced. Subjective health was evaluated based on responses from the survey (excellent, good, fair, poor, very bad, no information/no response) and was divided into two groups: good (excellent, good) and bad (fair, poor, very bad). BMI was divided into three groups: underweight (BMI < 18.5), overweight and obese (≥23), and normal weight. The comorbidities evaluated by doctors were assessed as hypertension, diabetes, dyslipidemia (DLP), stroke, and osteoarthritis (OA).

### 2.3. Assessment of Handgrip Strength

Since 2016, the KNHANES has measured HGS using a digital hand dynamometer (digital grip strength dynamometer, T.K.K 5401, Takei Scientific Instruments Co., Ltd., Tokyo, Japan) to evaluate muscle strength. Measurements were performed three times for each hand, and rest was taken for at least 60 s to recover muscle strength. The highest value was used as HGS. Poor HGS was defined as <28 kg for men and <18 kg for women, according to the 2019 definition of the AWGS [19].

### 2.4. Statistical Analyses

Because the KNHANES uses a complex multistage probability sample design, survey sample weights were considered when analyzing the KNHANES data, to produce unbiased cross-sectional estimates for the entire Korean population. To compare the demographics and health-related characteristics between the normal and poor grip strength groups, the chi-squared test for categorical variables and Student’s *t*-test for continuous variables were used. Univariable and multivariable logistic regression models were used to determine the association between SES and grip strength. To determine the interaction effect of household income or education and muscle exercise on grip strength, we performed a multivariable logistic regression analysis with an interaction term. The household income and muscle exercise interaction model was adjusted for sex, age, education, subjective health, aerobic exercise, and BMI, while the education and muscle exercise interaction model was adjusted for sex, age, household income, subjective health, aerobic exercise, and BMI. Three-level logistic regression analysis was performed using Baron and Kenny’s method to identify the mediating effects of muscle exercise on the relationships between household income, education level, and decline in grip strength [20]. Level 1 analyzed the relationship between the independent and mediating variables; level 2 analyzed the relevance between the independent and dependent variables; and level 3 analyzed the relevance between the independent and dependent variables and the relevance between the mediating and dependent variables when a mediating variable existed. Based on the results of levels 1 and 2, the Sobel test was performed to test the significance of the mediating effect [21]. All tests were two-tailed, and a *p*-value of <0.05 was considered statistically significant. Statistical analyses were performed using SPSS version 25 (IBM Corp., Armonk, NY, USA) and R (version 3.6.6; R Core Team, Vienna, Austria, 2019) [22].

## 3. Results

### 3.1. Demographic Characteristics of Participants According to HGS

Of the 6081 participants included in the study, 2688 (44.2%) were men and 3393 (55.79%) were women, with a mean age of 41.3 years. Of these, 1141 (15.7%) had a low income and 1116 (13.7%) had an elementary school education. Most of the participants (77.6%) did not perform muscle exercises. The distribution of the baseline characteristics according to grip strength is shown in Table 1. The prevalence of poor grip strength in this study was 13.7%. The mean ages of respondents with normal and poor HGS were 43.7 and 47.6, respectively. No difference was found in subjective health and aerobic exercise between the two groups; however, there were significant differences in sex, household income, education, muscle exercise, BMI, stroke, and OA between the two groups. The percentages of women in the normal and poor HGS groups were 46.5% and 75.5%, respectively. The percentages of participants in the lower quartile were 12.8% and 39.1% in the normal and poor HGS groups, respectively. In regard to education level, the percentages of participants who graduated only from elementary school were 10.0% and 44.6% in the normal and poor HGS groups, respectively. In the poor HGS group, 89.1% of the respondents reported no muscle exercise. The underweight ratios were 3.4% and 6.4% in the poor and normal HGS groups, respectively. There were differences in stroke and OA between the two groups; of the patients with poor grip strength, 6.0% were stroke patients and 20.9% were OA patients (Table 1).

Univariable analysis was performed to assess the effects of sex, age, household income, education, subjective health, muscle exercise, aerobic exercise, and BMI on poor HGS. It was revealed that the risk factors affecting poor HGS were female sex, old age, low household income, no muscle or aerobic exercise, underweight status, stroke, and OA (Table 2).

Furthermore, in the adjusted model, household income and education affected poor HGS. The odds ratio (OR) of poor HGS in 1 Quartile (1Q) was 1.637 (*p* = 0.005) compared to 4Q for household income. The OR of poor HGS in those with only elementary school education was 2.351 (*p* < 0.001) compared to those with a college/university education level. Muscle exercise was also a risk factor for poor HGS, and the OR for poor HGS was significantly higher at 1.526 (*p* = 0.013) in those who did not perform strength training than in those who had performed strength training in the past week (Table 2).

### 3.2. Interaction between SES and Muscle Exercise in Low SES Group 

Multivariable logistic regression analysis was performed to assess the interaction of muscle exercise with HGS according to the degree of household income and education status (Figure 2A). For each household income state, the OR for poor HGS decreased with muscle exercise. The OR of the lowest household income state without muscle exercise was 7.082 (*p* < 0.001) compared to the highest household income state with muscle exercise. The interaction model was adjusted for sex, age, education level, aerobic exercise, subjective health, BMI, and comorbidities (HTN, DM, DLP, stroke, and OA).

In each of the education levels, the OR for poor HGS decreased with muscle exercise (Figure 2B). In addition, the OR of the lowest education level (elementary school) group without muscle exercise was 14.711 (*p* < 0.001) compared to the highest education level (college/university) group with muscle exercise. The interaction model was adjusted for sex, age, household income, aerobic exercise, subjective health, BMI, and comorbidities (HTN, DM, DLP, stroke, and OS).

### 3.3. Mediating Effect of Muscle Exercise for HGS

A three-step logistic regression analysis using Baron and Kenny’s mediation method was performed to check the mediating effect of exercise on HGS. The significance of the three-step logistic regression analysis was confirmed using the Sobel test (Table 3A and Figure 3A,B). In step 1, the effect of household income (independent variable) on muscle exercise (dependent variable) was statistically significant in 1Q compared to 4Q (B = 0.532, *p* = 0.001). The effect of muscle exercise (dependent variable) on poor HGS was statistically significant. In step 2 (pathway c), the effect of household income (independent variable) on poor HGS (dependent variable) was statistically significant in 1Q compared to that in 4Q (B = 0.572, *p* = 0.001). In step 3 (pathway c’), the effect of household income on poor HGS when the mediating variable of muscle exercise was present was also statistically significant (B = 0.552, *p* = 0.002). All three steps were statistically significant, and the effect size of step 2 was larger than that of step 3 (OR from 1.772 to 1.736, z = 2.373, *p* = 0.017). Therefore, muscle exercise (a mediating variable) showed a partial mediating effect on the relationship between household income and poor HGS.

In addition, a three-step logistic regression analysis was performed for education level. In step 1 (pathway a), the effect of education level (independent variable) on muscle exercise (mediating variable) was statistically significant in the elementary school-level group compared to that in the college/university-level group (B = 0.494, *p* = 0.003). The effect of muscle exercise (mediating variable) on poor HGS (dependent variable) was statistically significant (pathway b). In step 2 (pathway c), the effect of education level (independent variable) on poor HGS (dependent variable) was statistically significant in the elementary school-level group compared to the college/university-level group (B = 0.862, *p* < 0.001). In step 3 (pathway c’), the effect of education level on poor HGS when the mediating variable of muscle exercise was present was also statistically significant (B = 0.837, *p* < 0.001). All three steps were statistically significant, and the effect size of step 2 was larger than that of step 3 (OR from 2.368 to 2.309, z = 2.489, *p* = 0.012). Therefore, muscle exercise (a mediating variable) showed a partial mediating effect on the relationship between education level and poor HGS. 

## 4. Discussion

In this study, low SES increased the risk of poor muscle strength by a factor of 1.6–2.3. Exercise improved muscle strength, and a difference was found in improving muscle strength by exercise depending on SES. Further confirmation may support this finding.

Low SES, which is known to cause adverse health outcomes through several pathways, is also an important risk factor for sarcopenia [14]. In the PURE study, involving 140,000 subjects and measuring the mean value of HGS according to low-, middle-, and high-income states, muscle strength was lower in the low-income state [23]. In the Louisiana Osteoporosis Study, the risk of sarcopenia was more than two times higher in the low-income state [24,25]. In a study using the KNHANES VI, muscle strength was decreased in the low education-level group, while in the Hallym Aging Study, muscle strength was decreased by a factor of more than 2.5 in the low education-level group [24,25]. In this study, the risk of poor HGS increased with both low household income and low education. This is consistent with previous findings showing that low SES reduces muscle strength and is a risk factor for sarcopenia. A lack of exercise would have a negative influence on muscle strength. A study using KNHANES VI showed low muscle strength when no exercise was performed [26], while a study by Miyazaki et al. showed similar results [17].

No drugs are effective for the treatment of sarcopenia. To date, exercise has been the best treatment, as it can increase muscle ability, capacity, and protein synthesis [27,28]. In a recent meta-analysis, some improvements in muscle strength were observed when exercise and dietary supplements were combined and managed [29]. Exercise has a positive influence on the neuromuscular system and improves muscle mass and strength by increasing hormone concentrations and protein synthesis rates [17,25,28]. However, people of low SES may have different perceptions of exercise necessity, nutrition, and activity compared with people of high SES [30]; hence, the effectiveness of improving muscle strength through exercise may be different. In this study, exercise further improved muscle strength in subjects of low SES.

Despite the many studies and policies in this area, the improvement of SES is not easy. The results of this study suggest that exercise is particularly important for the prevention of sarcopenia in patients of low SES. In reality, exercise is easier to modify and intervene in compared to other risk factors. Therefore, it is necessary to evaluate SES in people with low muscle strength suspected of sarcopenia, and, for those with a low SES, education on proper exercise may be required.

This study had several limitations. As this was a cross-sectional study, there was a limit to confirming the causal relationships between SES, muscle exercise, and muscle strength. Therefore, a longitudinal study is needed to confirm that exercise improves muscle strength. In addition, since this was a large-scale national survey, not a study using data that the researchers investigated for study purposes, only one item could confirm muscle exercise. Further studies that can supplement the specific measurement methods are needed. Because the implementation of exercise was checked using a survey, information biases may have occurred depending on the participants’ memory. We tried to reduce potential bias by evaluating whether or not muscle exercise was performed rather than the degree to which it was performed. However, further studies using specific recording methods, such as writing an exercise diary or measuring the amount of exercise with an accelerometer, are needed. Lastly, our study lacked information on nutritional status, which may affect muscle strength.

## 5. Conclusions

This study confirmed that exercise significantly improves muscle strength, especially in individuals with a low SES. Therefore, SES should be evaluated if sarcopenia is suspected. For patients with low SES, education on exercise is required.

## Figures and Tables

**Figure 1 healthcare-09-01244-f001:**
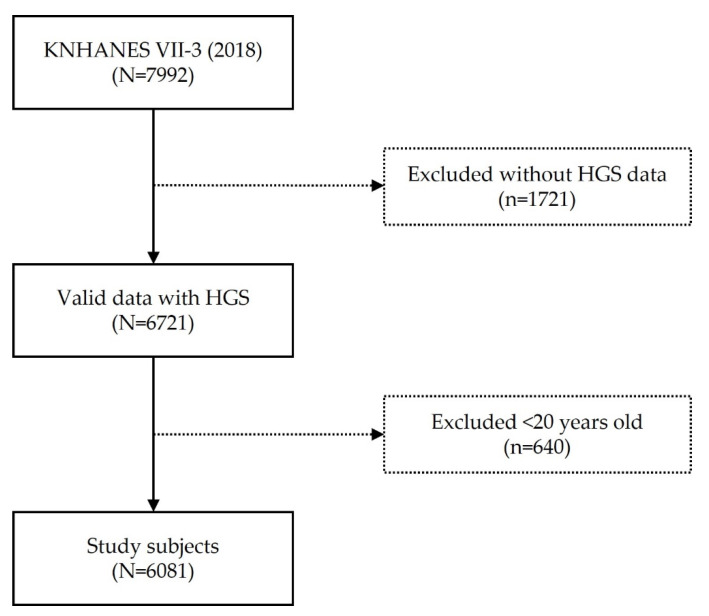
Flowchart.

**Figure 2 healthcare-09-01244-f002:**
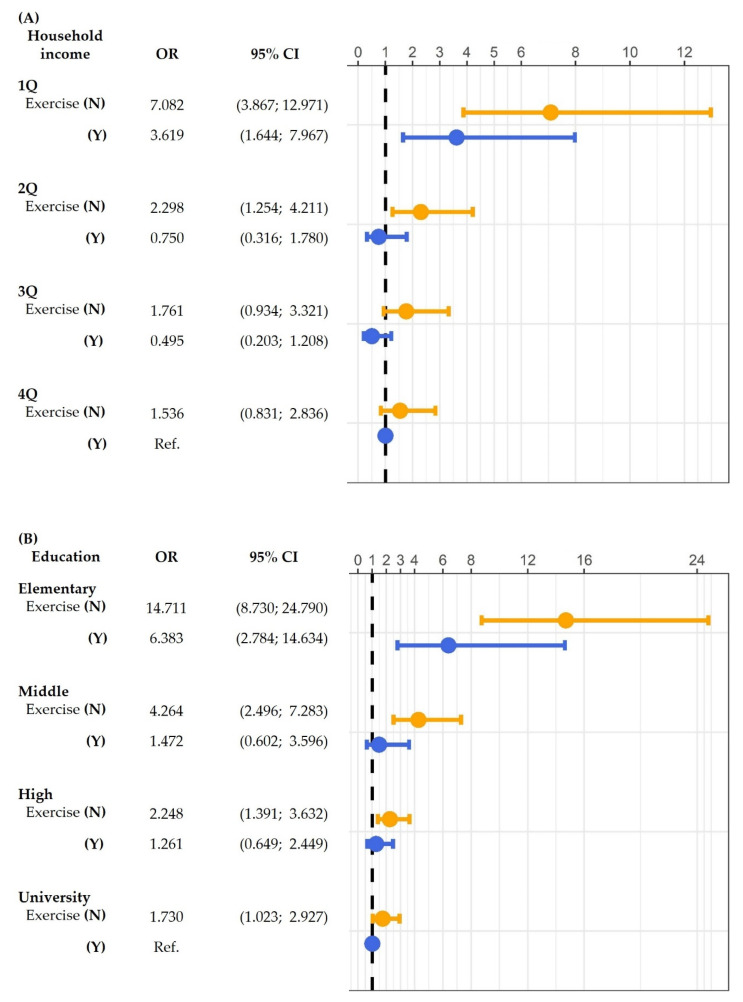
Interaction of (**A**) household income and (**B**) education with muscle exercise: odds ratio (OR) for poor HGS. *p*-values were calculated using multivariable logistic regression analysis. CI, confidence interval; Q, quartile; N, no; Y, yes; HGS, handgrip strength; Ref., reference.

**Figure 3 healthcare-09-01244-f003:**
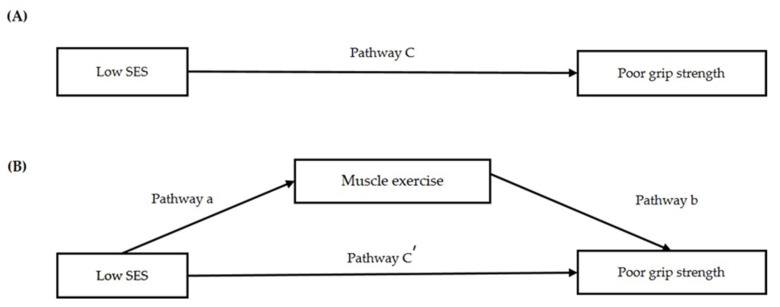
Regression coefficients for the association (**A**,**B**) between low SES and poor grip strength, as mediated by muscle exercise.

**Table 1 healthcare-09-01244-t001:** Demographic characteristics of the study participants according to grip strength.

Variables	Total (n = 6081)	Normal HGS (n = 5247)	Poor HGS (n = 834)	*p* ^a^
sex (female)	3393 (49.7)	2781 (46.5)	612 (75.5)	**<0.001**
age (year), mean (SD)	41.3 (0.5)	43.7 (0.4)	47.6 (1.4)	**<0.001 ^b^**
household income				**<0.001**
4Q (highest)	1806 (30.7)	1664 (32.1)	142 (19.6)	
3Q	1653 (29.0)	1512 (30.2)	141 (19.5)	
2Q	1462 (24.5)	1286 (24.9)	176 (21.7)	
1Q (lowest)	1141 (15.7)	770 (12.8)	371 (39.1)	
education				**<0.001**
college/university	2190 (41.3)	2056 (43.8)	134 (21.0)	
high school	1954 (36.5)	1793 (38.0)	161 (24.1)	
middle school	586 (8.4)	497 (8.2)	89 (10.2)	
elementary school	1116 (13.7)	720 (10.0)	396 (44.6)	
poor subjective health	4175 (69.2)	3525 (68.0)	650 (78.4)	0.484
no muscle exercise	4637 (77.6)	3930 (76.1)	707 (89.1)	**0.018**
no aerobic exercise	3405 (55.0)	2844 (53.3)	561 (68.4)	0.552
BMI (kg/m^2^)				**0.002**
underweight (<18.5)	210 (3.7)	169 (3.4)	41 (6.4)	
overweight and obese (≥23)	2103 (35.0)	1855 (35.6)	248 (30.5)	
HTN	1492 (19.6)	1131 (17.4)	361 (37.8)	0.141
DM	573 (7.4)	429 (6.4)	144 (14.8)	0.370
DLP	1155 (15.4)	930 (14.5)	225 (23.0)	0.073
stroke	146 (1.9)	92 (1.3)	54 (6.0)	**0.007**
OA	698 (8.2)	494 (6.6)	204 (20.9)	**0.018**

Abbreviations: SD, standard deviation; Q, quartile; BMI, body mass index; HGS, handgrip strength; HTN, hypertension; DM, diabetes mellitus; DLP, dyslipidemia; OA, osteoarthritis. Values are presented in numbers and percentages unless stated otherwise. ^a^, *p*-values were calculated using the chi-squared test. ^b^, *p*-values were calculated using Student’s *t*-test. Boldface indicates statistical significance (*p* < 0.05).

**Table 2 healthcare-09-01244-t002:** Univariable and multivariable logistic regression models: odds ratios for poor HGS.

Variables	Univariable				Multivariable			
	OR	95% Confidence Interval		*p* ^a^	OR	95% Confidence Interval		*p* ^a^
		Lower	Upper			Lower	Upper	
sex								
male	1				1			
female	3.537	2.903	4.309	**<0.001**	2.655	2.138	3.297	**<0.001**
age (year), mean (SD)	1.058	1.047	1.069	**<0.001**	1.029	1.016	1.041	**<0.001**
household income								
4Q (highest)	1				1			
3Q	1.056	0.767	1.454	0.736	0.921	0.653	1.298	0.635
2Q	1.427	1.083	1.880	**0.012**	0.943	0.697	1.277	0.704
1Q (lowest)	4.977	3.707	6.683	**<0.001**	1.637	1.159	2.312	**0.005**
education								
college/university	1				1			
high school	1.322	0.989	1.768	0.059	1.085	0.808	1.456	0.586
middle school	2.593	1.859	3.617	**<0.001**	1.125	0.761	1.622	0.553
elementary school	9.337	6.765	12.887	**<0.001**	2.351	1.629	3.394	**<0.001**
poor subjective health	1.708	1.345	2.169	<0.001	1.115	0.853	1.457	0.425
no muscle exercise	2.559	1.877	3.489	**<0.001**	1.526	1.094	2.129	**0.013**
no aerobic exercise	1.895	1.550	2.318	**<0.001**	1.093	0.881	1.355	0.416
BMI (kg/m^2^)								
normal weight	1				1			
underweight	1.823	1.211	2.744	0.059	2.645	1.678	4.168	**0.009**
overweight	0.826	0.677	1.007	**0.001**	0.731	0.578	0.924	**<0.001**
HTN	2.892	2.387	3.504	**<0.001**	1.137	0.904	1.429	0.271
DM	2.535	2.018	3.184	**<0.001**	1.182	0.899	1.554	0.231
DLP	1.768	1.439	2.171	**<0.001**	0.789	0.612	1.016	0.066
stroke	4.780	3.115	7.336	**<0.001**	2.236	1.352	3.699	**0.002**
OA	3.813	3.009	4.831	**0.001**	1.406	1.087	1.818	**0.010**

Abbreviations: OR, odds ratio; SD, standard deviation; Q, quartile; HGS, handgrip strength; BMI, body mass index (underweight, BMI < 18.5; overweight, BMI ≥ 23); HTN, hypertension; DM, diabetes mellitus; DLP, dyslipidemia; OA, osteoarthritis. ^a^ *p*-values were calculated using univariable logistic regression analysis. Boldface indicates statistical significance (*p* < 0.05).

**Table 3 healthcare-09-01244-t003:** Mediating effect of muscle exercise on the association between (A) low household income, (B) low education state, and poor HGS.

**(A) Steps (Pathway)**	**B**	**SE**	**OR**	**95% CI**		** *p* **	**Sobel Test**
				**Lower**	**Upper**		**z (*p*)**
Step 1 (a). I (1Q) → M	0.532	0.151	1.702	1.264	2.292	0.001	2.373 (0.017)
(b) M → G	−0.417	0.169	0.664	0.475	0.928	0.017	
Step 2 (c). I (1Q) → G	0.572	0.174	1.772	1.257	2.498	0.001	
Step 3 (c’). I (1Q) → G	0.552	0.172	1.736	1.235	2.440	0.002	
**(B) Steps (Pathway)**	**B**	**SE**	**OR**	**95% CI**		** *p* **	**Sobel Test**
				**Lower**	**Upper**		**z (*p*)**
Step 1 (a). E (1E) → M	0.494	0.165	1.639	1.184	2.271	0.003	2.489 (0.012)
(b) M → G	−0.407	0.172	0.665	0.474	0.935	0.019	
Step 2 (c). E (E) → G	0.862	0.187	2.368	1.635	3.429	<0.001	
Step 3 (c’). E (E) → G	0.837	0.187	2.309	1.596	3.342	<0.001	

B, estimate; SE, standard error; OR, odds ratio; CI, confidence interval; I (1Q), household income (1Q, reference: 4Q); E (E), education (elementary school, reference: college/university); M, muscle exercise; G, grip strength; HGS, handgrip strength; SES, socioeconomic status. Pathway a, the relationship between the independent and mediating variables; pathway b, the relevance between the mediating and dependent variables; pathway c, the relevance between the independent and dependent variables; pathway c’, the relevance between the independent and dependent variables. Pathways were calculated using multivariable logistic regression analysis.

## Data Availability

Publicly available datasets were analyzed in this study. The data can be found here: http://knhanes.cdc.go.kr/knhanes/sub03/sub03_01.do.

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
