# Peer review of "Muscle Exercise Mitigates the Negative Influence of Low Socioeconomic Status on the Lack of Muscle Strength: A Cross-Sectional Study"

_healthcare, 2021, doi:10.3390/healthcare9101244_

Round 1

Reviewer 1 Report

Thank you for the opportunity to review this manuscript. Some comments are relevant for the improvement and understanding of the manuscript.

- Please add to the discussion what these findings might imply in changes and clinical applicability.
- The conclusion must respond directly to the objective.

Reviewer 2 Report

The article is well written, also the contents are sufficiently novel and interesting for the current version. This work fills up the knowledge gaps of the exercise results in insufficient muscle strength improvement according to SES level which previous articles cannot address. The topic is original and necessary for the current literature. However, some areas of improvement are needed, especially in the discussion and language expression. My comments on the manuscript are below:

Title:

-The title describes the article clearly except for the word “sarcopenia”. Sarcopenia was defined consensus as "age-related loss of muscle mass, plus low muscle strength, and/or low physical performance" and specified cutoffs for each diagnostic component through the Asian Working Group for Sarcopenia (AWGS) 2014. The inclusion criteria of subjects above 20 years old in this paper are only muscle strength, so it is recommended to change “sarcopenia” to “lack of/insufficient muscle strength”, also as is the case elsewhere in the text.

Abstract:

-The abstract reflects the content of the article well.

Introduction

- In the introduction, the authors clearly stated that the risk of physical disability increased for the muscle strength decreased, and the hand grip strength(HGS) was chosen to measure the muscle strength. But the relationship among socioeconomic status(SES), HGS and exercise was not well presented.The study hopes to research the difference of HGS in different SES and the effect of exercise on HGS, so as to interpretation that exercise can improve the negative effect of low SES on muscle strength. The detailed presentation in exercise improving muscle strength and the difference of HGS in different SES brings new evidence according to my opinion. Please reorganize the introduction.

-Line 36, the abbreviations "ASM" in the formula should be explained.

-Line 44 to 48, to use “dual-energy X-ray absorptiometry” instead of “Dual-Energy X-ray absorptiometry”, “bioelectrical impedance analysis ” instead of “Bioelectrical impedance analysis ”. And the meaning of the letters "M" and "F" should be explained when they first appear.

-Line 53, in the sentence “In a longitudinal study … knee extensor strength.”, the highest physical disability incidence appeared the lowest quartile of knee extensor strength in men, but it appeared when the the highest quartile in women. Therefore, the gender differences should be taken into account.

-In the light of the information provided in the introduction, what did the author expect to find? Please add your purpose and hypothesis, but not only the statistical analysis method in the last paragraph of the introduction.

Method:

-Actually, the author has not accurately explain how the participant were recruited. Inclusion criteria and exclusion criteria should be clearly defined.

-Line 108, the BMI calculated formula should be kg/m2 but not kg/m2.

-Line 110, in the sentence”Participants were divided into two groups: no exercise at all and 1 day or 2 days more.”, “1 day or 2 days more” was indeterminate.

Results:

-The information described in this part could be obtained from the tables and figures, thus the expression should be more concisely.

-The participants should be stated uniformly throughout the text instead of “patients” in line 160 and “respondents” in line 168.

-Line 191, in the table note, the he table note of BMI is repeated.

-Line 194,the title of section 3.2 should be “Interaction Between SES and Muscle Exercise in Low HGS Group”.

Discussion:

-Line 260, the sentence “Low SES, which is … for sarcopenia[20]” was repeated with it in line 294.

-Line 271, what did the author want to express in the first sentence? It maybe lack of exercise would have negative influence on sarcopenia.

-Line 271 to 281, the relationship between exercise and SES was still not explained clearly.The significance and role of muscle exercise as a mediating variable should be fully described, which is the highlight of this study.

-Line 282, “… sarcopinia. dehydroepi- …”  should be “… sarcopinia. Dehydroepi- …”.

-Line 282 to 293, this paragraph proposed deletion due to the presentation of drugs did not improve muscle strength was inconsistent with the purpose of this study.

Conclusion:

-Line 314 to 315, “the rate of exercise for individuals with low SES is still low” was not discussed in the discussion section, thus could not get the conclusion of “policies that include appropriate educational programs and social support for exercise are necessary”.

Tables, Figures, Images:

-In the distribution of Table 1, when the total sample size was 6,081, the frequency and the relative frequency of women was 3,393 and 55.79%, respectively. However,it is not the case for other relative frequencies calculated based on this total sample. For example, the relative frequency for low income group sample size 1,141 of 6,081 could not be 15.7%. In addition, the same number of decimal places should be retained.

-In Table 2, what were the “lower” and “upper” respected for? The author only obtained HGS data, the above words could not represent the lower and upper limbs here. The results of the uni-variable and multi-variable logistic regression analysis were both listed in Table2, which one did the author refer to? The p-values more than 0.05 were also boldfaced for high school and underweight.  

Reviewer 3 Report

In this paper, the authors perform a series of statistical analysis to find the relationship between exercise, muscle strength and SES. However, the major contribution of this work to the literature is not significant. In the discussion, a variety of past studies were listed, but the novelty of this work, and the new insights brought to this field is not clearly stated.

In addition, a fundamental limitation of this work that caused my concern is the scarcity and accuracy of measurements. "Muscle exercise was evaluated as the percentage of strength exercises such as push-ups, sit-ups, dumbbells, weights, and iron bars practiced for more than two days during the previous week." This is by participant recall and to use a single data point to represent the participant entire living habits, which makes the results less scientifically reliable. Some subjective measures are also involved: "Subjective health was evaluated based on responses from the survey (excellent, good, fair, poor, very bad, no information/no response), and was divided into two groups: good (excellent, good) and bad (fair, poor, very bad)."  The study is largely subject to bias and errors, and the justification for grouping, specifically why "fair" was grouped to "bad" is confusing. Even though the authors lightly addressed this issue in row 301-308, this should be further improved.

Some additional changes needed:

  1. Please clarify the abbreviation for ASM on row 36
  2. Row 54-55 is not very clear: why women with highest knee extensor strength has a higher risk of physical disability?
  3. Capitalize first word in the sentence on row 282
  4. Repeated sentence: row 260, 294

Round 2

Reviewer 2 Report

- The author still insisted on discussing the effect of exercise on sarcopenia in the manuscript, but the survey participants selected by the author were not patients with sarcopenia, so the evidence was insufficient.

- Inconsistent sentence patterns are confusing. Did the author mean that no exercise group was for exercise for less than 1 days a week, and exercise group was forexercise more than 2 days a week?

- The number of sample sizes is still confusing. And as shown in Figure 1, the logic of 7992-640 = 6721,6721-640 = 6081 is also very confusing.

- In Table 2, numbers greater than 0.05 such as 0.059 and 0.066 are still bold to indicate the existence of differences, and corresponding modifications are not seen.
